# Specific Targeting of Antiapoptotic Bcl-2 Proteins as a Radiosensitizing Approach in Solid Tumors

**DOI:** 10.3390/ijms23147850

**Published:** 2022-07-16

**Authors:** Benjamin Sobol, Osama Azzam Nieto, Emily Lara Eberlein, Anna-Lena Scherr, Lars Ismail, Annika Kessler, Luisa Nader, Maximilian Schwab, Paula Hoffmeister, Nathalie Schmitt, Dirk Jäger, Stefan Welte, Katharina Seidensaal, Petros Christopoulos, Christoph Heilig, Katharina Kriegsmann, Stefan Fröhling, Mark Kriegsmann, Jochen Hess, Bruno Christian Köhler

**Affiliations:** 1Department of Medical Oncology, National Center for Tumor Diseases (NCT) Heidelberg, University Hospital Heidelberg, 69120 Heidelberg, Germany; benjaminsobol@hotmail.com (B.S.); osama.azzamnieto@googlemail.com (O.A.N.); 19imc10690@fh-krems.ac.at (E.L.E.); anna-lena.scherr@nct-heidelberg.de (A.-L.S.); lars.ismail1986@googlemail.com (L.I.); annika.kessler@t-online.de (A.K.); luisa.nader@gmx.de (L.N.); mce.schwab@gmail.com (M.S.); paula.hoffmeister@googlemail.com (P.H.); nathalie.schmitt@nct-heidelberg.de (N.S.); dirk.jaeger@nct-heidelberg.de (D.J.); petros.christopoulos@gmail.com (P.C.); 2German Cancer Consortium (DKTK), 69120 Heidelberg, Germany; christoph.heilig@nct-heidelberg.de (C.H.); stefan.froehling@nct-heidelberg.de (S.F.); 3Department of Radiation Oncology, Heidelberg University Hospital, 69120 Heidelberg, Germany; stefan.welte@med.uni-heidelberg.de (S.W.); katharina.seidensaal@med.uni-heidelberg.de (K.S.); 4Translational Lung Research Center, German Center for Lung Research (DZL) at Heidelberg University Hospital, 69120 Heidelberg, Germany; 5Department of Translational Medical Oncology, National Center for Tumor Diseases (NCT) Heidelberg, University Hospital Heidelberg, 69120 Heidelberg, Germany; 6Department of Hematology, Oncology and Rheumatology, University of Heidelberg, 69120 Heidelberg, Germany; katharina.kriegsmann@med.uni-heidelberg.de; 7Institute of Pathology, University of Heidelberg, 69120 Heidelberg, Germany; mark.kriegsmann@med.uni-heidelberg.de; 8Molecular Mechanisms of Head and Neck Tumors, German Cancer Research Center (DKFZ), 69120 Heidelberg, Germany; jochen.hess@med.uni-heidelberg.de; 9Department of Otorhinolaryngology, Head and Neck Surgery, Heidelberg University Hospital, 69120 Heidelberg, Germany

**Keywords:** Bcl-2, Mc-1, Bcl-x_L_, radiotherapy, therapy resistance, cell death, synovial sarcoma, head and neck squamous cell carcinoma, non-small-cell lung cancer

## Abstract

Avoidance of therapy-induced apoptosis is a hallmark of acquired resistance towards radiotherapy. Thus, breaking resistance still challenges modern cancer therapy. The Bcl-2 protein family is known for its regulatory role in apoptosis signaling, making Bcl-2, Mcl-1 and Bcl-x_L_ promising targets. This study evaluates the effects of highly specific inhibitors for Bcl-x_L_ (WEHI-539), Bcl-2 (ABT-199) and Mcl-1 (S63845) as radiosensitizers. Covering a broad spectrum of solid tumors, Non-Small-Cell Lung Cancer (NSCLC), Head and Neck Squamous Cell Carcinoma (HNSCC) and synovial sarcoma cell lines were exposed to fractionated radiation as standard therapy with or without Bcl-2 protein inhibition. Protein expression was detected by Western blot and cell death was assessed by flow cytometry measuring apoptosis. In contrast to NSCLC, a high level of Bcl-x_L_ and its upregulation during radiotherapy indicated radioresistance in HNSCC and synovial sarcoma. Radioresistant cell lines across all entities benefited synergistically from combined therapy with Bcl-x_L_ inhibition and fractionated radiation. In NSCLC cell lines, Mcl-1 inhibition significantly augmented radiotherapy independent of the expression level. Our data suggest that among antiapoptotic Bcl-2 proteins, targeting Bcl-x_L_ may break resistance to radiation in HNSCC, synovial sarcoma and NSCLC in vitro. In NSCLC, Mcl-1 might be a promising target that needs further investigation.

## 1. Introduction

The Bcl-2 protein family regulates a cell’s fate by keeping a balance between pro- and anti-apoptotic proteins. The proapoptotic family members (BAX, BAK) react to apoptosis inducing signals by permeabilization of the outer mitochondrial membrane followed by subsequent cell death execution. BH3-only proteins (BIM, PUMA, BAD) operate proapoptosis by binding antiapoptotic proteins [1,2]. Bcl-2, Bcl-x_L_ and Mcl-1 are the most prominent representatives of the antiapoptotic branch of the family. They inhibit apoptosis by sequestering their proapoptotic relatives—a mechanism that is widely exploited by malignant cells to survive. Thus, resisting cell death is a hallmark of cancer harboring therapeutic implications [3].

Following this deliberation, inhibitors of antiapoptotic Bcl-2 family members (BH3 mimetics) were developed. The selective Bcl-2 inhibitor Venetoclax (ABT-199) was the first compound approved for treatment of chronic lymphocytic leukemia (CLL) and certain types of acute myeloid leukemia (AML) [4,5,6]. Specific inhibitors for Bcl-x_L_ (e.g., WEHI-539) and Mcl-1 (e.g., S63845) have been evaluated in preclinical studies and are currently entering early clinical investigation. Promising results for solid tumors in terms of safety and efficiency from early clinical data will lead to further trials [7,8,9].

Radiotherapy is one of the cornerstones of many cancer therapies. Approximately 60% of patients with solid tumors receive an either curative or palliative irradiation within their treatment [10]. By dose fractionation, tumor-cell killing may be augmented [11]. Yet, the radiation-induced tumor cell death is countered by the development of resistance mechanisms, limiting the capabilities of treatment, and originating the urge for new, efficient strategies with fewer side effects. In this context, targeting of antiapoptotic Bcl-2 proteins may open a new therapeutic window.

The effect of ionizing radiation has been described in many studies. Amongst various forms of cell death, radiation-induced apoptosis is the most important effect [12,13]. Ionizing radiation causes lethal lesions such as DNA double-strand breaks, thus activating a series of cellular damage responses [14]. The inability of sufficient repair triggers the intrinsic apoptotic pathway which is ultimately regulated by the antiapoptotic Bcl-2 proteins [15,16,17]. By upregulation of Bcl-2 proteins, the cell may retort to cellular stress. Inhibiting these control mechanisms could be radiosensitizing in solid tumors.

As for many other tumor types, modern radiotherapy is a fundamental part of (multimodal) treatment strategies in non-small-cell lung cancer (NSCLC), head and neck squamous cell carcinoma (HNSCC) and soft tissue sarcomas such as synovial sarcoma. These tumor types may represent the variety of tumor histologies targeted by radiotherapy [18,19,20]. The aim of this study is to broadly evaluate the potential of BH3 mimetics as radiosensitizers in NSCLC, HNSCC and synovial sarcoma. By detecting changes in cellular expression patterns during irradiation, cytotoxic effects of BH3 mimetics and the combination with standard-of-care therapeutic irradiation, we establish new potential therapeutic options.

## 2. Results

### 2.1. Ionizing Radiation and Its Effects on NSCLC, HNSCC and Synovial Sarcoma

Cell death was assessed by flow cytometry via DNA fragmentation after using dose-fractionation schemes ranging from 4 × 0 to 4 × 5 Gy on NSCLC, HNSCC and synovial sarcoma cells (Figure 1). Radiation induced cell death in all cell lines; however, sensitivity to radiation strongly varied across entities and cell lines. NSCLC cell lines stood out for being resistant, and apart from H1975 the other three cell lines (H1650, H838, A549) did not surpass the mark of 10% cell death at the highest fractionation of 4 × 5 Gy (Figure 1A). In contrast, HNSCC cell lines reacted increasingly to elevated radiation levels, with Detroit-562 being more resistant, followed by Cal-27 and then FaDu (Figure 1A). Synovial sarcoma cell lines revealed very heterogenous results, with SW982 cells showing almost no measurable apoptosis while Fuji and SYO-1 were comparably most affected across all cell lines (Figure 1A).

We then validated our findings by Western blotting for phosphorylated H2A histone family member X (pH2AX) and cleaved Poly-ADP-ribose polymerases (cleaved PARP) at 4 × 0 Gy and 4 × 2 Gy; cells were harvested 24 h after the last radiation session. pH2AX was used to measure additional double-strand DNA damage induced by radiation. Excluding SW982 (synovial sarcoma) and A549 (NSCLC), all other cell lines showed a detectable increase in pH2AX due to radiation. In line with the shown FACS data, there were no or barely detectable cleaved PARP in those cell lines described as being resistant above confirming very low apoptosis rates (Figure 1B).

Furthermore, we wanted to explore if radiation induces changes in Bcl-2 protein expression level and if the basal protein expression could predict sensitivity to radiation. Amongst NSCLC cell lines, no relevant regulatory effects could be detected. Additionally, basal protein levels did not affect cell death with heterogenous expression patterns within the Bcl-2 family, leading to similar results (H1650, H838, A549) (Figure 1B). Interestingly, HNSCC (Detroit-562) and synovial sarcoma (SW982) cell lines expressing the highest levels of Bcl-x_L_ were most resistant to radiation. The data also indicate that for these cell lines, upregulation could play a role in escaping radiation-mediated cell death, though densitometric analysis showed no significance in upregulation. Bcl-2 and Mcl-1 appear to play a minor role with expression levels being low, and apart from a slight upregulation of Bcl-2 among HNSCC cell lines, no detectable up- or down-regulation (Figure 1B).

### 2.2. Higher Expression Levels of Basal Bcl-x_L_ Negatively Correlate with Radiation-Induced Cell Death in HNSCC and Synovial Sarcoma Cells

In contrast to NSCLC cell lines, Bcl-x_L_ seemed to play a role in radiation resistance among HNSCC and synovial sarcoma cell lines. Thus, a Western blot including all six cell lines was performed to adequately compare basal expression levels. Relative Bcl-x_L_ expression was measured by densitometric analysis as described above and plotted against delta cell death at 4 × 4 Gy. The coefficient of determination (R^2^, r^2^) was calculated to measure the closeness of the data to the fitted linear regression (Figure 2A). Within the panel consisting of HNSCC and synovial sarcoma cell lines (*n* = 6), a correlation of R^2^ = −0.78 (Figure 2A) was calculated, while NSCLC cell lines showed no correlation with R^2^ = −0.16 (Figure 2A). The same analysis was performed for Bcl-2 and Mcl-1 without significant correlation.

### 2.3. Selective Inhibition of the Bcl-2 Protein Family Induces Apoptosis in NSCLC, HNSCC and Synovial Sarcoma Cell Lines

Inhibitor concentrations of 0.5, 1, 5 and 20 µM of the BH3 mimetics WEHI-539 (Bcl-x_L_), ABT-199 (Bcl-2) and S63845 (Mcl-1) were used to inhibit the prosurvival Bcl-2 protein family. Dose titration results were depicted graphically as a heat map (Figure 2B) showing a heterogenous reaction to molecular inhibition. To summarize, ABT-199 has the lowest impact on cell death among all entities excluding SYO-1 and Fuji cell lines. In contrast, WEHI-539 and S63845 were able to induce higher levels of cell death across entities although a correlation between expression levels and efficacy of BH3 mimetics could not be shown. Detailed analyses of induced cell death are shown in Appendix A.

### 2.4. Mcl-1 and Bcl-x_L_ Inhibition Can Lead to Higher Radiation-Induced Cytotoxicity among NSCLC Cell Lines

Subsequently, we investigated potential radiosensitizing effects of BH3 mimetics in an experimental setup as seen in Figure 3A. Cell lines were treated with an inhibitor dose that should not surpass delta cell death >10%. Then, a total of four dose fractions that should induce delta 5–15% cell death were applied, interrupted by a splitting process on day 3 to evade radiation resistance due to overgrowing cells. Radiation schedules and inhibitor concentrations were chosen according to the previously conducted experiments.

Apart from the cell line H1650, Mcl-1 and Bcl-x_L_ inhibition led to clear sensitization effects while Bcl-2 inhibition via ABT-199 played a negligible role in NSCLC cells. H1975 was irradiated with 4 × 2 Gy and treated with inhibitor concentrations according to the previously set conditions, showing an obvious susceptibility to Mcl-1 and Bcl-x_L_ inhibition (Figure 3B, *p* ≤ 0.001 for WEHI-539 and S63845). H838, a more resistant cell line, underwent a radiation schedule of 4 ×3 Gy, while A549, the most resistant NSCLC cell line, was irradiated with 4 × 5 Gy. Interestingly, H838 expressed the highest levels of Mcl-1 (Figure 1B) and showed a more profound cell death induction with 1 µM S63845 (*p* ≤ 0.01, Figure 3B), whereas A549 expressed the highest levels of Bcl-x_L_ (Figure 1B and Figure 2A) and showed a clear increase in radiation-induced damage when treated with 5 µM WEHI-539 (*p* ≤ 0.01, Figure 3B).

### 2.5. Bcl-x_L_ Inhibition Plays a Key Role in Radiosensitization among the Resistant HNSCC and Synovial Sarcoma Cell Lines

Amongst HNSCC, combined therapy at 4 × 1 Gy showed no significant effects with Cal-27 cells (Figure 4A). FaDu cells underwent a radiation schedule of 4 × 2 Gy and showed a slight sensitization effect with 20 µM ABT-199 and 1 µM S63845 (*p* ≤ 0.05, Figure 4A). A sharp increase could be demonstrated at 1 µM WEHI-539 (*p* ≤ 0.001, Figure 4A), although FaDu cells expressed the second lowest basal Bcl-x_L_ levels (Figure 1B and Figure 2A). Detroit-562 cells expressed the highest level of Bcl-x_L_ among HNSCC cells and were exposed to a radiation schedule of 4 × 2 Gy. Even at 20 µM dosing, ABT-199 and S63845 showed no effect (Figure 4A), while the WEHI-539 dose of 1 µM showed a significant radiation-enhancing effect (*p* ≤ 0.05, Figure 4A). As radiation schedule and inhibitor concentration of WEHI-539 did not match predetermined requirements for Detroit-562 cells, a panel consisting of WEHI-539 concentrations of 0.5, 0.75 and 1 µM was repeated with an adapted radiation schedule of 4 × 3 Gy, now showing clear radiosensitizing effects for the three concentrations (*p* ≤ 0.01, Appendix A).

Synovial sarcoma cells SYO-1 were irradiated with 4 × 0.5 Gy and showed a minimal increase in cell death with 0.075 µM WEHI-539, whereas Fuji cells were irradiated with 4 × 1 Gy and depicted a small sensitization effect across all BH3 mimetics (*p* ≤ 0.05, Figure 4B). Strikingly, the most resistant cell line SW982 showed a major increase in radiation-induced damage at 4 × 5 Gy while being treated with 0.5 µM of WEHI-539. Mean cell death at 4 × 5 Gy was 4.2%, with the control group being at 1.8%. The inhibitor alone induced 9% cell death. Combined therapy led to an impressive increase in apoptotic cells to 31.2% (*p* ≤ 0.001, Figure 4B). To conclude, our observations indicate a clear benefit of Bcl-x_L_ inhibition for radiotherapy in HNSCC and synovial sarcoma cell lines, especially in cell lines expressing high Bcl-x_L_ levels and showing Bcl-x_L_ upregulation under irradiation.

The above-stated results of Section 2.4 and Section 2.5. were graphically depicted in a heat map shown in Figure 5. Our data unravel an impressive potential of Bcl-x_L_, and specifically for NSCLC, Mcl-1 inhibition as a radiosensitizing approach in biologically different solid tumors. Synergism was defined as a combinatory effect surpassing the additive effect of both components.

## 3. Discussion

The objective of this study was to evaluate the effects of specific inhibitors for Bcl-x_L_ (WEHI-539), Bcl-2 (ABT-199) and Mcl-1 (S63845) in the context of fractionated photon irradiation in exemplary solid tumors. We investigated a soft-tissue sarcoma subtype (synovial sarcoma) as well as two different epithelial tumors (NSCLC and HNSCC), tumor types in which irradiation is an indispensable part of therapeutic concepts. Our HNSCC cell lines showed Human Papilloma Virus (HPV) negativity, and thus would correlate, in a clinical context, with a worse prognosis due to elevated resistance to radiotherapy and chemotherapy [21]. This is attributed to tumor protein 53 (TP53) mutation which could be found in our three cell lines, thus presenting a spectrum of more resistant cell lines that represent the majority of HNSCC. Similar to our findings, in NSCLC cell lines with EGFR mutations, Lu et al. discovered an upregulation of Bcl-2 and Bcl-x_L_ as a mechanism to escape cell death via tyrosine kinase inhibitors and sensitized them via BH3 mimetics [22]. Interestingly, while a p53 gene mutation correlated with poor survival in NSCLC patients, sole Bcl-2 abnormalities showed no significant influence on the prognosis, consorting with our findings that isolated Bcl-2 inhibition had no impact on our NSCLC cell lines’ vitality [23].

Even though radiotherapy is a mainstay in treatment for different stages of synovial sarcoma, HNSCC and NSCLC, repair mechanisms and resistances still compromise the outcome. To this day, radiotherapy protocols are complemented by conventional cytotoxic agents such as Cisplatin in HNSCC and NSCLC to overcome the above-mentioned obstacles [24,25]. The Bcl-2 family presents itself as one of the largest gene families to be affected by somatic copy-number alterations. Beroukhim et al. emphasize a major role of Mcl-1 and Bcl-x_L_ in cancer survival and even demonstrate that cancer cells with focal amplifications of *MCL1* and *BCL2L1*, the gene encoding Bcl-x_L_, especially depend on Mcl-1 and Bcl-x_L_ for growth and survival, making both proteins ideal targets of molecular inhibition [26].

Our findings revealed HNSCC and synovial sarcoma cell lines with high Bcl-x_L_ expression levels to be more resistant to irradiation and to increase Bcl-x_L_ expression under radiotherapy. Those results file into existing discoveries from other studies. HNSCC cell lines with high Bcl-x_L_ expression demonstrated an inferior response to chemotherapy or ionizing radiation and even turned out to be associated with poor clinical outcomes [27,28]. Ow et al. analyzed patients´ data and performed in vitro experiments that suggested Bcl-x_L_ expression as a major factor for treatment response, while resistance to radiation may be associated with high Mcl-1 levels [29]. Even though co-treatment of radiation with Bcl-2 inhibition showed no relevant effect in our experiments across all entities, sole Bcl-2 inhibition interestingly induced measurable cell death within the synovial sarcoma cell lines Fuji and SYO-1 and not among SW982. An explanation could be SW982 missing the pathognomonic SYT-SSX translocation which has been shown to directly regulate Bcl-2 expression among synovial sarcomas. Fairchild, C.K. et al. demonstrated a severe downregulation of Bcl-2 expression via knockdown of the mentioned fusion oncogene [30]. Coinciding with our findings, sole Bcl-2 inhibition among solid tumors has been shown to have a lesser impact than among hematopoietic malignancies, where Bcl-2 seems to have a pivotal role due to a high Bcl-2 dependency through sequestering of the proapoptic protein BIM. The sequestering can be annulled by competitive binding of Bcl-2 antagonists [31]. Bcl-2 inhibition has been broadly studied among estrogen-receptor-positive breast cancer due to its high Bcl-2 expression. Findings revealed that Bcl-2 inhibition increases cell death only when combined with other treatments such as endocrine therapy, mTOR inhibition or chemotherapy by priming the Bcl-2 protein due to higher binding of BIM. Other studies depicted some promising results among HNSCC when adding an inhibition of EGFR in cell lines with specific mutation patterns [32]. Therefore, Bcl-2 inhibition might be a target with a different combination scheme other than (fractionated) radiation.

The effect of Mcl-1 inhibition in NSCLC with concomitant radiotherapy warrants further attention. Other studies have shown that Mcl-1 is found to be a major regulator in avoiding apoptosis in NSCLC [33]. More precisely, cells with enhanced levels of Mcl-1 were more resistant to cytotoxic agents or ionizing radiation, while an inhibition of Mcl-1 increased the cytotoxic effect [34]. Other findings showed that Mcl-1 not only influences cancer survival via cytosolic regulation of intrinsic apoptosis but also plays an important role in DNA damage response by facilitating repair of DNA damage directly at the site of damage [35]. In addition, especially with radiation-induced double-strand breaks (DSBs) Mcl-1 seems to impede chromosomic aberrations by promoting DSB repair via homologous recombination [36]. A cells’ dependence on Mcl-1 seemed to have an inverse correlation with Bcl-x_L_ level, meaning that a high Mcl-1-to-Bcl-x_L_ mRNA ratio determines whether the cells entrust their survival to Mcl-1. All NSCLC cell lines we used showed a relatively identical high Mcl-1 expression. In a comparison of adenocarcinomas of the lung, Mcl-1 turned out to secure its survival and to be critical for its development [37]. Munkhbaatar et al. found high gains in Mcl-1 resulting after loss of functional TP53, indicating its advantageous role against chromosomal instability.

Sole treatment with WEHI-539 does not strongly affect the viability of Bcl-x_L_ high SW982. In combination with radiotherapy, Bcl-x_L_ inhibition induces massive cell death presumably due to insufficient DNA repair mechanisms. For synovial sarcoma, Barrott et al. already suggested the in vitro therapeutic potential of Bcl-x_L_ inhibition, underlining our findings [38].

Toxicity is a limiting factor for Bcl-2 inhibitor application. The combined Bcl-2/Bcl-x_L_ inhibitor ABT-263 showed in clinical trials dose-limiting side effects with thrombocytopenia, attributed to inhibition of Bcl-x_L_, and leukocytopenia, due to Bcl-2 inhibition [39,40,41]. After treatment with WEHI-539 in mouse models, low levels of erythrocytes and hemoglobin were observed [42]. We were able to use lower doses by combining Bcl-x_L_ inhibition with radiotherapy, thereby reducing the probability of adverse effects while still managing to obtain a notable increase in cell death. Our in vitro approach might be exploited in vivo, ultimately leading to clinical translation.

In conclusion, this study further unveils the capabilities of BH3 mimetics in solid tumors. The antiapoptotic Bcl-2 protein family seems to play an important role in ensuring survival in a tumor-type-specific pattern. Exclusively for NSCLC, Mcl-1 inhibition should be considered as radiosensitizer in follow-up studies. Applicable to all tumor types investigated here, Bcl-x_L_ inhibition needs further attention and may harbor great advantages for possible treatment strategies in the future in combination with radiotherapy.

## 4. Material and Methods

### 4.1. Cell Lines and Reagents

All human NSCLC and synovial sarcoma cell lines were purchased via ATCC (Manassas, VA, USA), and HNSCC cell lines via CLS (Cell Lines Service, Eppelheim, Germany). The following cell lines were used: H838, A549, H1650 and H1975 (NSCLC). SW982, Fuji and SYO-1 (Synovial sarcoma). Detroit-562, FaDu and Cal-27 (HNSCC). All NSCLC cell lines are adenocarcinomas. H1975 has mutant Tumor protein 53 (TP53) (R273H) and the Epidermal growth factor receptor (EGFR) has L858R/T790M double mutation while H1650 has wild-type p53 and EGFR exon 19 deletion (del E746-A750) [43]. Cell line A549 is known to have a homozygous mutation in the pro-oncogene Kirsten rat sarcoma (KRAS) (c34G>A/pG12S) [44]. The cell line H838 shows no driver mutation. Cell lines Fuji and SYO-1 show the chromosomal translocation t(X;18) (p11.2;q11.2) commonly found in synovial sarcomas that consists of a fusion of gene synovial sarcoma X (SSX) on Chromosome X and SS18 (Chromosome 18). Cell line SW982 is atypical, showing no translocation [45]. All HNSCC cell lines are Human papillomavirus (HPV)-negative and express TP53 mutation. Detroit-562 is of pharyngeal origin and shows, additionally to TP53, a Phosphatidylinositol-4,5-Bisphosphate 3-Kinase Catalytic Subunit Alpha (PIK3CA) mutation. FaDu, of hypopharyngeal origin, further expresses mutations for FAT atypical cadherin 1 (FAT1) and lysine demethylase 6A (KDM6A) and Cal-27, a tongue SCC, in addition, is Caspase 8 (CASP8)-, Mother against decapentaplegic homolog 4 (SMAD4)- and KDM6A-mutated [46]. Cells were grown and maintained at 37 °C and 5% CO_2_ in a humid atmosphere and cultured in Gibco’s RPMI 1640 Medium (Thermo Fisher, Schwerte, Germany) supplemented with 10% fetal calf serum (PAA Laboratories, Cölbe, Germany) and 1% Penicillin/Streptomycin (PAA). Solely A549 was grown and maintained in supplemented Gibco’s F-12K (Kaighn’s) Medium (Thermo Fisher, Schwerte, Germany). Selective BH3 mimetics ABT-199 (Venetoclax; Bcl-2-inhibitor) was provided by Abbvie, S63845 (Mcl-1-inhibitor) was purchased from Selleckchem (Munich, Germany) and WEHI-539 (Bcl-x_L_-inhibitor) from Hycultec (Beutelsbach, Germany).

### 4.2. Protein Isolation, SDS-Page, Densitometry and Western Blotting

Cells were seeded as indicated for protein analysis. Protein isolation was performed and lysates separated using a 12% acrylamide concentration for the separating gel. After separation and solubilization via Sodium dodecyl sulfate–Polyacrylamide gel electrophoresis (SDS-PAGE), the samples were blotted onto a nitrocellulose membrane using a wet blot transfer apparatus (Bio-Rad, Hercules, CA, USA) following standard procedures. Immunodetection was performed with following primary antibodies: Bcl-x_L_ (# 2764, Cell Signaling Technology, Danvers, MA, USA), Mcl-1 (# sc-819, Santa Cruz Biotechnology, Heidelberg, Germany), Bcl-2 (# ab692, Abcam, Cambridge, UK), Cleaved PARP (# 5625, Cell Signaling Technology, Danvers, MA, USA), pH2AX (# 9718, Cell Signaling Technology, Danvers, MA, USA), Tubulin (# T8203, Sigma–Aldrich, St. Luis, MO, USA), and Actin (# sc-1616, Santa Cruz Biotechnology, Heidelberg, Germany). Secondary antibodies (Santa Cruz Biotechnology, Heidelberg, Germany) were detected using PerkinElmer’s (Rodgau, Germany) Enhanced Chemiluminescence Substrate (# NEL103001EA), and signal intensity was measured with ImageJ^®^ (by Wayne Rasband at NIH, Bethesda, MD, USA). Densitometric analysis was then performed by norming the expression band to their respective loading control. The value of 1.0 was assigned to a cell line of reference and densitometric values were plotted against the amount of induced cell death at 4 × 4 Gy.

### 4.3. Cell-Death Analysis by Flow Cytometry

Cells were treated according to protocol before the supernatant and cells were transferred to the corresponding FACS tubes. Accutase™ (Thermo Fisher, Schwerte, Germany) was used for cell detachment prior to flow cytometry to reduce potential proteolytic effects of trypsinization. After centrifugation, the supernatant was discarded and the cell pellet resuspended in buffer consisting of 50 µg/mL propidium iodide (P4864, Sigma-Aldrich, St. Luis, MO, USA), 0.1% (*w*/*v*) sodium citrate and 0.1% (*v*/*v*) Triton X-100. After a 24 h incubation period at 4 °C in the dark, cell death was measured following the protocol of Nicoletti et al. using a CANTO II flow cytometer (Becton Dickinson [BD], Franklin Lakes, NJ, USA) [47]. Cell cycle analysis was performed using FACS Diva 6 (BD) and FlowJo 7.6.5. (Tree Star Inc., Ashland, OR, USA). Hypodiploid PI-stained cells were considered apoptotic.

### 4.4. Photon-Beam Radiotherapy for In Vitro Experiments

In vitro experiments with ionizing radiation were performed with the X-Rad 320 x-Ray system cabinet from Precision X-Ray (Madison, CT, USA), parameters were set to 1 Gy/min at 320 kV and 12.5 mA for all experiments. A focus-to-skin distance (FSD) of 50 cm was chosen with the adjustable sample shelf and photon beam filtration was achieved using a filter consisting of 1.5 mm aluminum, 0.25 mm copper and 0.75 mm tin. Radiation dosage was applied with a fractionated schedule of 4 consecutive days of radiation with 1 to 5 Gy in each session to mimic clinical partitioning of the overall dose to smaller fractions. Cells were seeded 24 h prior to radiation to reach a confluency of 70–80% and split after the second radiation session to circumvent overgrowth. The harvesting point was chosen 24 h after the last session. Figures depicting the experimental setup were created using BioRender.com (accessed on 14 July 2022).

### 4.5. Statistical Analysis

Statistical analysis was performed using GraphPad PRISM 9.0 (GraphPad Software, San Diego, CA, USA). In vitro experiments were executed in biological triplicates and summarized as mean values with standard deviation. Normally distributed data were analyzed using an unpaired t-test, and for comparisons of more than two variables a two-way analysis of variance (two-way ANOVA) matching both factors was conducted. Correlations were verified using linear regression models. Statistical significance is indicated as * *p* < 0.05; ** *p* < 0.01; *** *p* < 0.001.

## Figures and Tables

**Figure 1 ijms-23-07850-f001:**
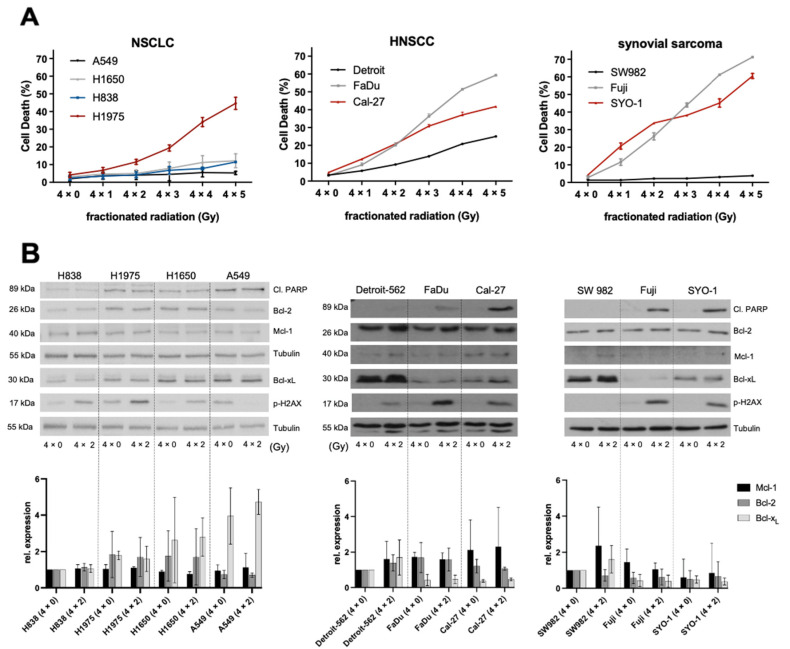
Effects of fractionated radiation on NSCLC, HNSCC and synovial sarcoma cell lines. (**A**) FACS analysis of cell death induction during fractionated radiation. Experiments were performed in triplicate and cell death was measured as mean ± SD. The depicted FACS analyses are representative of at least three independent experiments. (**B**) Representative Western blot analysis of protein development during fractionated radiation and basal protein expression of the Bcl-2-family with Tubulin serving as loading control. Below, a densitometric analysis was performed by norming the expression bands to their respective loading control and using for each entity one cell line as reference. Relative expression was compared between three independent experiments and measured as mean ± SD.

**Figure 2 ijms-23-07850-f002:**
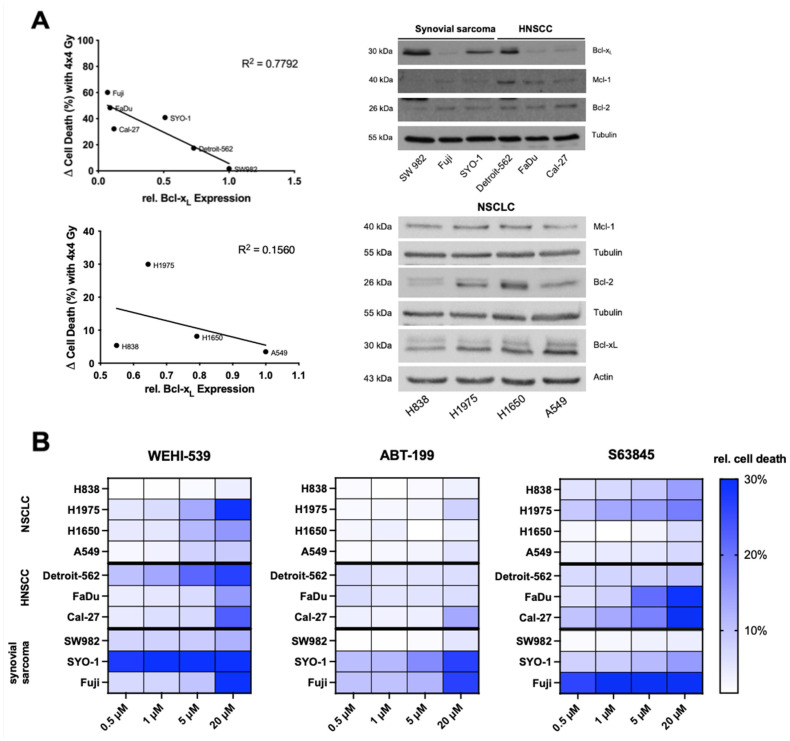
Correlation of relative Bcl-x_L_ expression with radiation induced cell death. (**A**) The coefficient of determination (R^2^) was calculated by plotting the results of a densitometric analysis of shown Western blots against radiation-induced cell death at delta 4 × 4 Gy. Representative Western blots; Tubulin/Actin served as loading control. HNSCC and synovial sarcoma cell lines were grouped together due to their similar reaction to radiation with Bcl-x_L_ upregulation. (**B**) Graphical depiction via heatmap of inhibitor-induced cell death. FACS analysis of cell death induction during treatment with BH3 mimetics in different concentrations. Experiments were performed in triplicate and cell death was measured as mean ± SD. Findings are depicted graphically in a heat map for overview purposes. Results are representative of at least three independent experiments.

**Figure 3 ijms-23-07850-f003:**
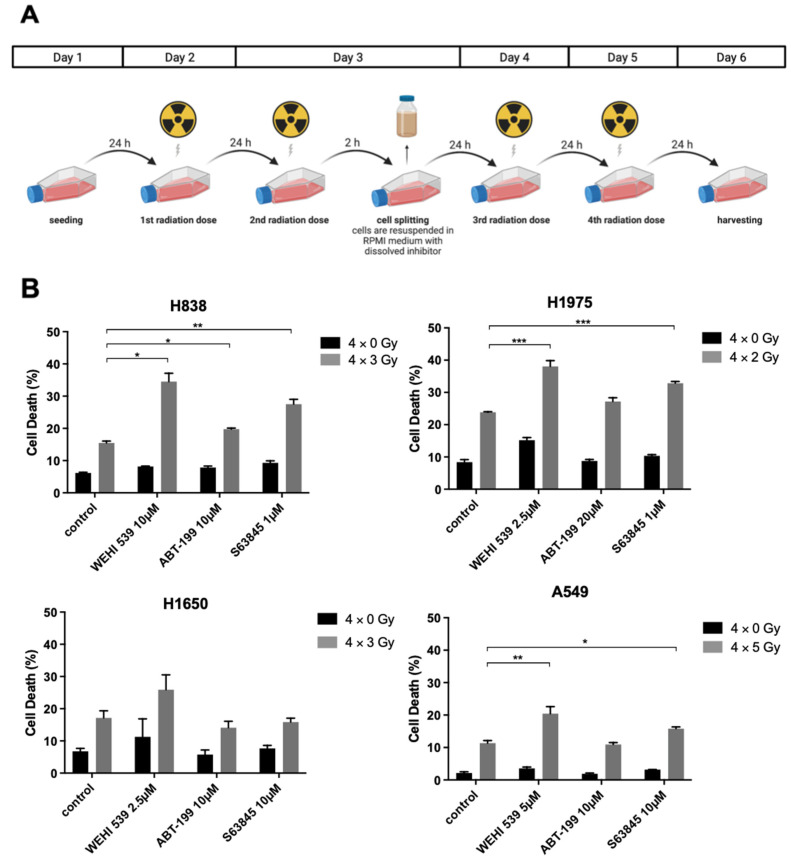
Set up for in vitro experiments and radiosensitization results of NSCLC cell lines (**A**) Cell lines undergoing a fractioned radiation schedule during molecular inhibition with BH3 mimetics. (**B**) Combination of irradiation and molecular inhibition in NSCLC cell lines and FACS analysis of induced cell death. Experiments were performed in triplicate and cell death was measured as mean ± SD. The depicted FACS analyses are representative of at least three independent experiments. * *p* ≤ 0.05, ** *p* ≤ 0.01, *** *p* ≤ 0.001.

**Figure 4 ijms-23-07850-f004:**
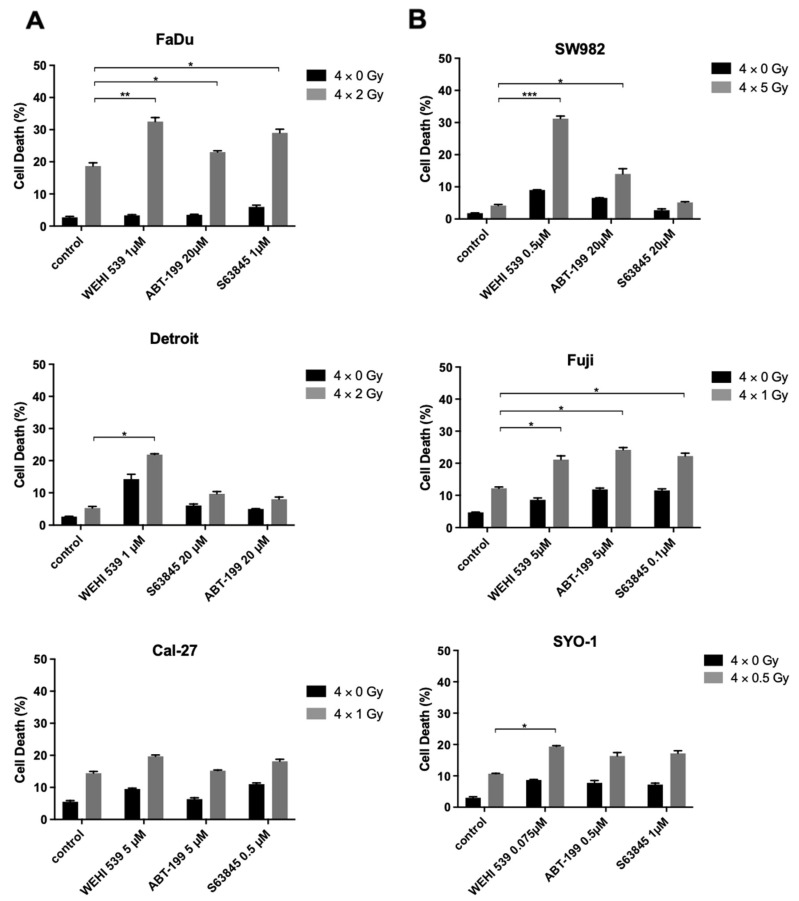
Radiosensitization results of HNSCC and synovial sarcoma cell lines Combination of irradiation and molecular inhibition in (**A**) HNSCC and (**B**) synovial sarcoma cell lines measured by FACS analysis. Experiments were performed in triplicate and cell death was measured as mean ± SD. The depicted FACS analyses are representative of at least three independent experiments. * *p* ≤ 0.05, ** *p* ≤ 0.01, *** *p* ≤ 0.001.

**Figure 5 ijms-23-07850-f005:**
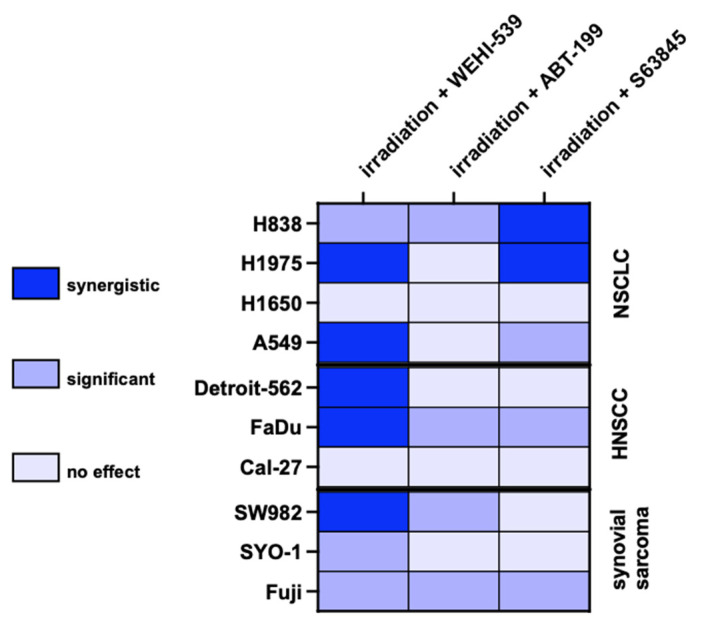
Graphical overview of NSCLC, HNSCC and synovial sarcoma cell lines undergoing combined fractioned radiation and BH3-inhibition as depicted in Figure 3B and Figure 4. The label synergistic was defined as the combinatory effect surpassing the additive effect of irradiation and molecular inhibition.

## Data Availability

All data generated or analyzed during this study are included in this published article and its Appendix A.

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
