# Peer review of "Specific Targeting of Antiapoptotic Bcl-2 Proteins as a Radiosensitizing Approach in Solid Tumors"

_ijms, 2022, doi:10.3390/ijms23147850_

Round 1
Reviewer 1 Report
The ms by Sobol et al. presents an interesting study on the use of specific BCl-2 protein family inhibitors (BH3 mimetics), as promising therapeutic options in NSCLC, HNSCC and synovial sarcoma treatments. The authors describe a role of such inhibitors as radiosensitizers through a fractioned radiation schedule clearly described in the text, and performed on a panel of tumor cell lines representing NSCLC, HNSCC and synovial sarcoma.
The use of cell lines as drug-response preclinical model suitable for clinical samples, is more than an issue. In fact, our limited understanding of the similarities and differences between cell lines and patient tumors remains a key challenge for translating findings from cell lines to the clinic.
In that sense, more details on the cell lines used here should be provided and discussed by the authors, to evaluate how much these cell lines can match with the tumor types described.
Overall, the manuscript is well outlined but it provides preliminary results on BcL-2 family proteins expression upon radiation in the different tumor context. It is very hard to draw conclusion on Bcl-2 family protein expression and radiation sensitivity in only one WB experiment, and in absence of the normalization and statistical analysis.
Therefore, it is mandatory to provide the normalization and statistical analysis of all western blotting data, and to add them in the revised version.
Below are specific points for the authors:
1) Fig. 1B - Please provide Bcl-2, Bcl-xL and Mcl-1 normalization on corresponding control and represent the results with histograms and statistics to evaluate either the higher expression of Bcl-2 in Detroit cells as stated in lines 181- 182, or the effects of fractionated radiation. It seems that in NSCLC cells panel less protein amount was loaded compare to other cell lines
2) Please align WB panel of Fig. 1B below the proper tumor type
3) Line 192: where it was described the densitometric analysis?
4) Fig 2A: right panel - add the tumor type below WB
5) I suggest to add 3.3 section to 3.2 and to provide a biological explanation or possible mechanism on the apoptotic role of WEHI-539 and S63845 in absence of a correlation with Bcl-2.
6) Please align the significance upper the proper bar in fig 3B and Fig 4 A , B
Reviewer 2 Report
The well conceived manuscript presenting a clear, comprehensive and well-structured study of relevance to the field and of interest to the scientific community. The introduction provides sufficient background. The cited references are current and the data are interpreted appropriately. The results are presented clearly and corroborated by appropriate graphics. The discussion and statements are coherent and supported by the listed citations. The conclusions are supported by the results.
I recommend the manuscript to be accepted in present form.
Round 2
Reviewer 1 Report
The paper has been improved and can be accepted in its present form